# Fast evolution of SOS-independent multi-drug resistance in bacteria

Le Zhang[1†], Yunpeng Guan[1†], YuenYee Cheng[1], Nural N Cokcetin[2], Amy L Bottomley[2], Andrew Robinson[3], Elizabeth J Harry[2], Antoine M van Oijen[3], Qian Peter Su[1,4]*, Dayong Jin[1]*

[1]Institute for Biomedical Materials and Devices (IBMD), University of Technology Sydney, Ultimo, Australia; [2]Australian Institute for Microbiology & Infection (AIMI), University of Technology Sydney, Ultimo, Australia; [3]School of Chemistry and Molecular Bioscience, University of Wollongong, Wollongong, Australia; [4]School of Biomedical Engineering, University of Technology Sydney, Ultimo, Australia

*For correspondence:
Qian.Su@uts.edu.au (QPS);
dayong.jin@uts.edu.au (DJ)

†These authors contributed equally to this work

Competing interest: The authors declare that no competing interests exist.

## eLife Assessment

This study presents a **valuable** observation of how deletion of a major repair protein in bacteria can facilitate the rise of mutations that confer resistance against a range of different antibiotics. The data presented are **convincing**, and the authors addressed the concerns raised by the reviewers in their resubmission, improving the strength of their findings.

**Abstract** The killing mechanism of many antibiotics involves the induction of DNA damage, either directly or indirectly, which activates the SOS response. RecA, the master regulator of the SOS response, has been shown to play a central role in the evolution of resistance to fluoroquinolones, even after short-term exposure. While this paradigm is well established for DNA-damaging antibiotics, it remains unclear whether β-lactams elicit similar resistance dynamics or depend on RecA and SOS-mediated mechanisms. In this study, we observed a rapid and stable evolution of β-lactam resistance (20-fold MIC increase within 8 hr) in *Escherichia coli* lacking RecA after a single exposure to ampicillin. Contrary to expectation, this resistance emerged through an SOS-independent mechanism involving two distinct evolutionary forces: increased mutational supply and antibiotic-driven selection. Specifically, we found that RecA deletion impaired DNA repair and downregulated base excision repair pathways, while concurrently repressing the transcription of antioxidative defence genes. This dual impairment led to excessive accumulation of reactive oxygen species (ROS), which in turn promoted the emergence of resistance-conferring mutations. While ampicillin treatment did not alter survival, it selectively enriched for rare mutants arising in the RecA-deficient and ROS-elevated background. Collectively, our findings demonstrate that this oxidative environment, together with compromised DNA repair capacity, increases genetic instability and creates a selective landscape favouring the expansion of resistant clones. These results highlight the repair-redox axis as a key determinant of bacterial evolvability under antimicrobial stress.

## Introduction

Addressing bacterial infections caused by emerging and drug-resistant pathogens represents a major global health priority. Bactericidal antibiotics can exert their effects on cells by directly or indirectly causing DNA damage or triggering the production of highly destructive hydroxyl radicals (*Rebecca, 2015*; *Kohanski et al., 2007*; *Baquero and Levin, 2021*). This, in turn, initiates a protective mechanism known as the SOS response, which enables bacterial survival against the lethal impacts of

antibiotics by activating intrinsic pathways for DNA repair (*Žgur-Bertok, 2013*; *Wigley, 2013*; *Harms et al., 2016*; *Maslowska et al., 2019*). The activation of DNA repair processes relies on specific genes, such as *recA*, which encodes a recombinase involved in DNA repair, and *lexA*, a repressor of the SOS response that can be inactivated by RecA (*Cox, 2007*).

Studies have demonstrated that a single exposure to fluoroquinolones, a type of antibiotic that induces DNA breaks and triggers the SOS response, leads to the development of bacterial resistance in *Escherichia coli* through a RecA and SOS response-dependent mechanism (*Barrett et al., 2019*). Given the crucial role of RecA in the SOS response, inhibiting RecA activity to deactivate the SOS response presents an appealing strategy for preventing the evolution of bacterial resistance to antibiotics (*Pavlopoulou, 2018*). Similarly, exposure to fluoroquinolones induces the SOS response and mutagenesis in *Pseudomonas aeruginosa*, and the deletion of *recA* in this pathogen results in a significant reduction in resistance to fluoroquinolones (*Mercolino et al., 2022*).

Unlike fluoroquinolones, β-lactam antibiotics induce a RecA-dependent SOS response in *E. coli* through impaired cell wall synthesis, mediated by the DpiBA two-component signal system (*Miller et al., 2004*). The development of antibiotic resistance, triggered by exposure to β-lactams, has been extensively investigated using the cyclic adaptive laboratory evolution (ALE) method. Mutations that arise during cyclic ALE experiments are attributed to errors occurring during continued growth, necessitating multiple rounds of β-lactam exposure to drive the evolution of resistance in *E. coli* cells (*Jahn et al., 2017*; *Levin-Reisman et al., 2017*). However, the precise roles of RecA and SOS responses in the development of resistance under short-term β-lactam antibiotics exposure remain unclear.

Recently, there has been a growing interest in understanding the impact of the stress-induced accumulation of reactive oxygen species (ROS) on bacterial cells (*Dwyer et al., 2007*). While exploring methods to harness ROS-mediated killing has the potential to enhance the effectiveness of various antibiotics (*Luan et al., 2018*; *Zhao and Drlica, 2014*; *Zhao et al., 2015*), the role of ROS in antimicrobial activity has become a topic of controversy following challenges to the initial observations (*Keren et al., 2013*; *Imlay, 2015*). The generation of ROS has been found to contribute to the development of multidrug resistance, as prolonged exposure to antibiotics in cyclic ALE experiments is known to generate ROS, leading to DNA damage and increased mutagenesis (*Kohanski et al., 2010*; *Takahashi et al., 2017*). Nevertheless, there is still limited knowledge regarding the consequences of ROS accumulation in bacteria when the activity of RecA or the SOS response is suppressed.

Here, we report that a single exposure to β-lactam antibiotics can rapidly drive the evolution of multidrug resistance in *E. coli* lacking RecA. This process reflects a two-step evolutionary mechanism: RecA deficiency increases mutational supply by impairing DNA repair, repressing antioxidant gene expression, and promoting ROS accumulation; subsequently, antibiotic pressure selectively enriches resistant variants from this hypermutable population.

## Results

### Single β-lactam exposure accelerates resistance evolution in the *recA* mutant strain through a selection-driven mechanism

To investigate the impact of the SOS response on bacterial evolution towards β-lactam resistance, we generated a *recA* mutant strain (*ΔrecA*) from the *E. coli* MG1655 strain. Initially, we conducted an ALE experiment using a slightly modified treatment protocol (*Andersson and Hughes, 2014*) on the wild type and *ΔrecA* strains. During a period of three weeks, the cells were subjected to cycles of ampicillin exposure for 4.5 hr at a concentration of 50 µg/mL (10 times the MIC) each day (*Figure 1—figure supplement 1A*; *Fridman et al., 2014*). As anticipated based on previous studies, the intermittent ampicillin treatments over the course of three weeks resulted in the evolution of antibiotic resistance in the wild type strain (*Figure 1—figure supplement 1B*). However, a significantly accelerated development of resistance was found in the *ΔrecA* strain, with the average time to resistance reduced to 2 days (*Figure 1—figure supplement 1B*). More importantly, we observed that resistance even emerged in the *ΔrecA* strain after a single exposure for 8 hr to ampicillin (*Figure 1A–C*). Meanwhile, after 8 hr of treatment with 50 µg/mL ampicillin, the survival rates of both wild type and *ΔrecA* strain were consistent (*Figure 1—figure supplement 2*). To ensure that the emergence of resistance we observed was not illusory due to technical issues during the *recA* knockout process, we employed another *ΔrecA* strain (JW2669-1) provided by the Coli Genetic Stock Centre (CGSC) with the same

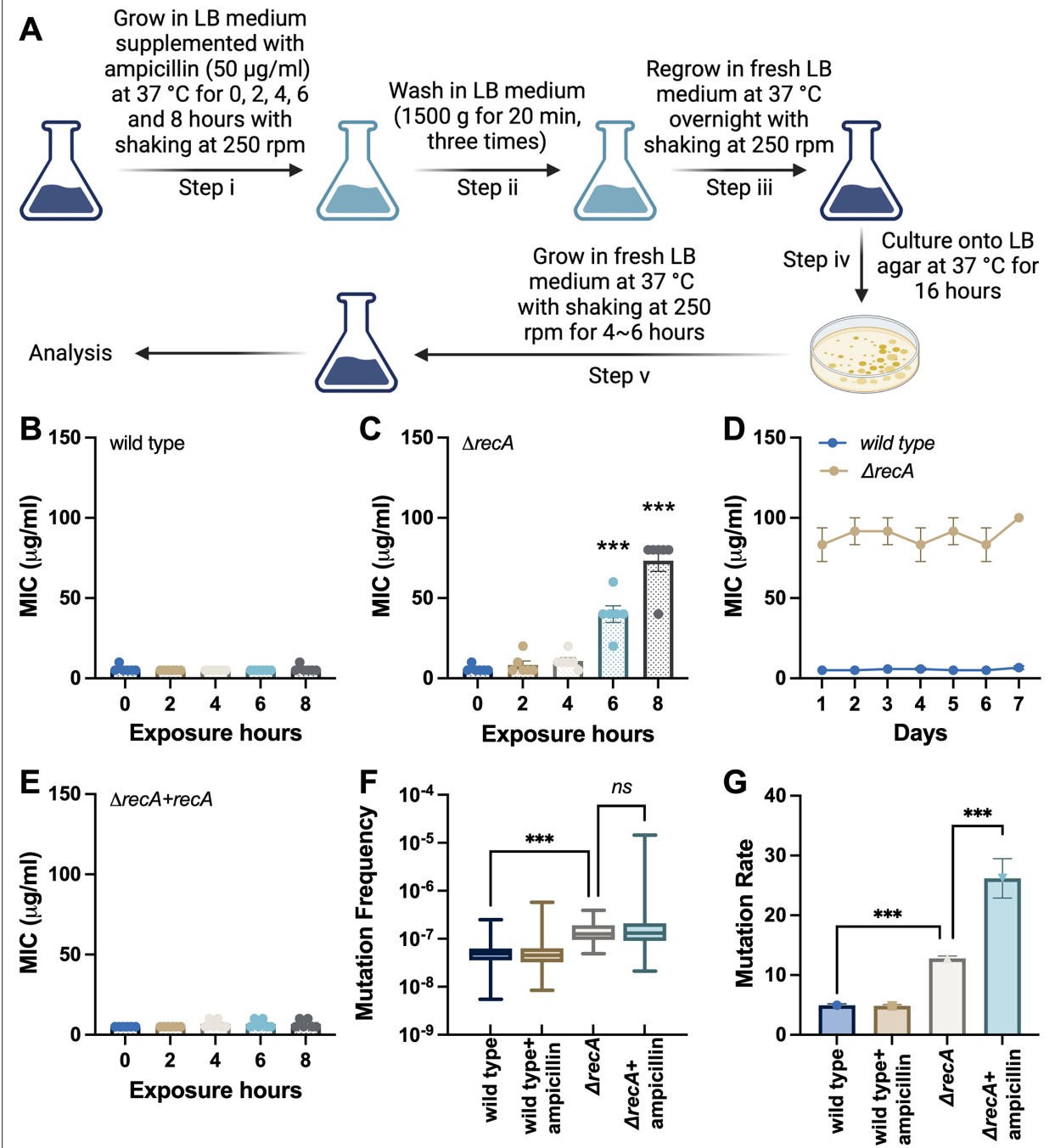

**Figure 1.** Fast evolution of antibiotic resistance in *E. coli* recA mutant strain. (**A**) Experimental flow for the single exposures to antibiotics in *E. coli* strains. Step i: an overnight culture (1×10⁹ CFU/mL cells) was diluted 1:50 into 30 mL LB medium supplemented with 50 µg/mL ampicillin and incubated at 37 °C with shaking at 250 rpm for 0, 2, 4, 6, and 8 hr; Step ii: after each treatment, the ampicillin-containing medium was removed by washing twice in a fresh LB medium; Step iii: the surviving isolates were resuspended in 30 mL fresh LB medium and regrown overnight at 37 °C with shaking at 250 rpm; Step iv: cell cultures were plated onto LB agar and incubated for 16 hr at 37 °C; Step v: single colonies were inoculated in 30 mL fresh LB medium and cultured at 37 °C with shaking at 250 rpm for 4–6 hr. (**B**) MICs of ampicillin were measured against the wild type *E. coli* strain after single exposures to ampicillin. (**C**) MICs of ampicillin were measured against the Δ*recA* strain after single exposures to ampicillin. (**D**) After the treatment of Step v, cells were continuously cultured in an antibiotic-free medium for seven days. MICs of ampicillin were measured each day. (**E**) MICs of ampicillin were measured against the Δ*recA* strain treated with ampicillin, where the expression of RecA was restored using plasmid-based constitutive expression of *recA* before the treatment of Step i. (**F**) Distribution of rifampicin-resistant mutant counts following single β-lactam exposure (96 replicate cultures in each group). Statistical comparison of mutation frequency (median values) used the Kruskal-Wallis test followed by Dunn's multiple comparisons. (**G**) Mutation rate (mutations per culture) estimates derived by maximum likelihood analysis. Each experiment was independently repeated at least six times using parallel

*Figure 1 continued on next page*

*Figure 1 continued*

replicates, and the data are shown as mean ± SEM. Significant differences among different treatment groups are analysed by independent t-test, *p<0.05, **p<0.01, ***p<0.001, *ns*, no significance.

The online version of this article includes the following figure supplement(s) for figure 1:

**Figure supplement 1.** Long-term exposures to ampicillin induced the evolution of resistance in the wild type and Δ*recA E. coli* strain.

**Figure supplement 2.** The survival rate after a single exposure to ampicillin at 50 µg/mL for 0, 2, 4, 6, and 8 hr in the wild type and Δ*recA* strain.

**Figure supplement 3.** Single exposures to ampicillin induced the evolution of resistance in the Δ*recA*$^{CGSC}$ strain (JW2669-1).

**Figure supplement 4.** Single exposures to other β-lactam antibiotics induced the evolution of resistance in the Δ*recA* strain.

**Figure supplement 5.** Distribution fitting and fluctuation analysis support a selection-driven resistance mechanism in the Δ*recA* strain.

killing procedure. The results from both bacterial strains were consistent (*Figure 1—figure supplement 3A and B*). To further investigate, we treated both the wild type and Δ*recA* cells with other β-lactams, including penicillin G and carbenicillin, at concentrations equivalent to 10 times the MIC (1 mg/mL and 200 µg/mL, respectively) for 8 hours (*Reimer et al., 1981*; *Jones et al., 1979*). Consistently, these treatments also led to a fast evolution of antibiotic resistance in the Δ*recA* strain (*Figure 1—figure supplement 4A and B*).

To assess the stability of this accelerated antibiotic resistance acquired by the Δ*recA* strain, we conducted a study wherein the Δ*recA*-resistant isolates, originating from the initial 8 hr treatment with ampicillin, were continuously cultivated in a medium devoid of antibiotics for a period of seven days. Our findings revealed that once resistance was established, resistance remained stable and was able to be passed on to subsequent generations even in the absence of ampicillin (*Figure 1D*). Moreover, we performed a complementation experiment by introducing a plasmid containing *recA* under its native promoter into the Δ*recA* strain prior to Step i in *Figure 1A*, that is, before exposing the cells to ampicillin. Interestingly, this complemented strain displayed a comparable MIC to the isogenic wild type strain and maintained its sensitivity even after ampicillin treatment for up to 8 hr (*Figure 1E*).

Antibiotic resistance evolves through the combined effects of mutational events and selection imposed by antimicrobial pressure (*Baquero et al., 2021*). To clarify which mechanism underlies the rapid emergence of resistance in the Δ*recA* strain, we systematically analysed the mutation frequency and distribution patterns in response to ampicillin treatment. *Figure 1F* shows the distribution of rifampicin-resistant colony-forming units (CFUs) across 96 independent cultures of the wild type and Δ*recA* strains, with and without exposure to ampicillin (50 µg/mL for 8 hr). Although the treatment of ampicillin in the Δ*recA* strain displayed a highly skewed distribution with apparent jackpot cultures, non-parametric statistical comparison (Kruskal-Wallis with Dunn's test) did not detect a significant difference in overall mutation frequency compared to untreated Δ*recA*. This suggests that while the median mutation burden remained stable, rare resistant outliers were significantly enriched, which was a pattern consistent with selection rather than broad mutagenesis (*Luria and Delbrück, 1943*). Further, using the maximum likelihood estimation (MLE; *Zheng, 1999*), we calculated the mutation rates (mutations per culture) for each group (*Figure 1G*). While the Δ*recA* group exhibited a modest increase in baseline mutation rate compared to the wild type strain, the addition of ampicillin led to a significant increase in estimated mutation rate.

To further assess the statistical properties of these distributions, we fitted the observed data to a Poisson model (*Figure 1—figure supplement 5A*). The mutation frequency distributions in the wild type strain with or without the treatment of ampicillin conformed well to Poisson expectations, consistent with random spontaneous mutation events. In contrast, the single exposure to ampicillin significantly deviated from the Poisson model, indicating a non-random clonal enrichment process in the Δ*recA* strain (*Sarkar et al., 1992*). Finally, we applied a fluctuation test-based inference framework grounded in the Luria-Delbrück model to distinguish between mutation induction and clonal selection (*Foster, 2007*). As shown in *Figure 1—figure supplement 5B*, only the Δ*recA* strain treated with ampicillin exhibited a markedly non-Poisson distribution of mutant counts, characterised by a long right-skewed tail and the emergence of jackpot cultures. This distribution is inconsistent with uniform mutagenesis and instead supports a model in which antibiotic treatment selectively enriches rare early-arising resistant subpopulations. Together, these findings demonstrate that a single exposure to β-lactam antibiotics promotes a rapid and heritable evolution of antibiotic resistance in the Δ*recA* strain, predominantly through selection rather than de novo induction of mutations.

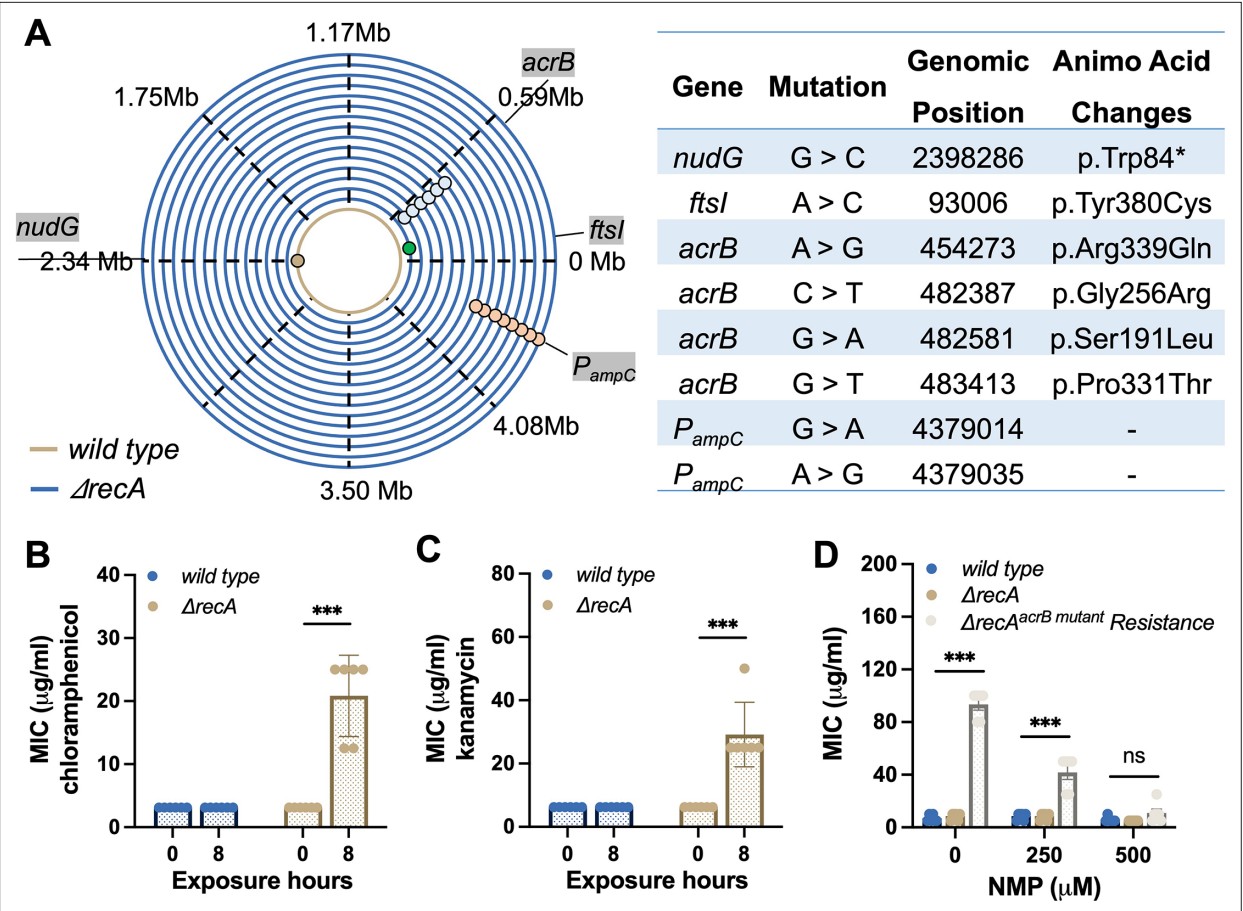

**Figure 2.** Rapid induction of drug resistance associated DNA mutations in the *recA* mutant strain. (**A**) Detection of drug resistance associated DNA mutations in the wild type and *ΔrecA* strain after the single exposures to ampicillin at 50 μg/mL for 8 hr. (**B**) MICs of chloramphenicol against the wild type and *ΔrecA* strain after the single treatment with ampicillin were tested. (**C**) MICs of kanamycin against the wild type and *ΔrecA* strain after the single treatment with ampicillin were tested. (**D**) The wild type, *ΔrecA*, and *ΔrecA*^acrB mutant^-resistant isolates were incubated with NMP at different concentrations for 12 hr. Subsequently, MICs were tested in these strains for resistance to ampicillin. Each experiment was independently repeated at least six times, and the data is shown as mean ± SEM. Significant differences among different treatment groups are analysed by independent t-test, *p<0.05, **p<0.01, ***p<0.001, *ns*, no significance.

The online version of this article includes the following figure supplement(s) for figure 2:

**Figure supplement 1.** The activity of β-lactamase was increased in the *ΔrecA* culture supernatants.

## Detection of drug resistance associated DNA mutations in *recA* mutant-resistant isolates

In bacteria, resistance to most antibiotics requires the accumulation of drug resistance associated DNA mutations that can arise stochastically and, under stress conditions, become enriched through selection over time to confer high levels of resistance (*Toprak et al., 2011*). Having observed a non-random and right-skewed distribution of mutation frequencies in *ΔrecA* isolates following ampicillin exposure, we next sought to determine whether specific resistance-conferring mutations were enriched in *ΔrecA* isolates following antibiotic exposure. We thereby randomly selected 15 colonies on non-selected LB agar plates from the wild type surviving isolates, and antibiotic screening plates containing 50 μg/mL ampicillin from the *ΔrecA*-resistant isolates, respectively, and performed whole-genome sequencing. We found that drug resistance-associated mutations were present in all resistant isolates, including the mutations in the promoter of the β-lactamase *ampC* (*P_ampC*) in eight isolates, the ampicillin-binding target PBP3 (*ftsI*) in one isolate, and the AcrAB-TolC subunit AcrB (*acrB*) mutations in six isolates (*Figure 2A*). A mutation in gene *nudG* was detected in wild type surviving isolates after the single

exposure to ampicillin (*Figure 2A*), which is involved in pyrimidine (d)NTP hydrolysis to avoid DNA damage (*Kamiya et al., 2001*). Other mutations were listed in *Supplementary file 1*.

The presence of $P_{ampC}$ mutations was accompanied by a significant increase in the production of β-lactamase in bacteria (*Figure 2—figure supplement 1*). This leads to specific resistance to β-lactam antibiotics. The gene *acrB* codes for a sub-component of the AcrAB-TolC multi-drug efflux pump, which is central in Gram-negative bacteria (*El Meouche and Dunlop, 2018*; *Blair et al., 2015*). Mutations in AcrAB-TolC enhance the efflux of antibiotics and confer resistance to multiple drugs (*El Meouche and Dunlop, 2018*). Consequently, after short-term exposure to ampicillin, the *ΔrecA* isolates carrying the *acrB* mutations exhibited resistance to other types of antibiotics, such as chloramphenicol and kanamycin (*Figure 2B and C*). Treatment with high concentrations of 1-(1-Naphthylmethyl) pipera-zine (NMP), an efflux pump inhibitor (EPI) that competitively blocks TolC-composed efflux pumps, successfully restored the sensitivity of *ΔrecA*-resistant isolates to ampicillin, bringing them back to the equivalent concentrations found in the wild type (*Figure 2D*). Collectively, these results suggest that β-lactam treatment rapidly selects for resistance-conferring mutations, which were enriched in *ΔrecA* isolates following short-term exposure.

## Hindrance of SOS-independent DNA repair in *recA*-mutant-resistant isolates

Impairment of DNA damage repair can accelerate the accumulation of mutations and influence bacte-rial adaptability under antibiotic stress. While RecA is best known for regulating the SOS response, we asked whether its absence also impacts resistance evolution through SOS-dependent or -inde-pendent repair mechanisms. To investigate it, we first tested the ability of various mutants involved in different pathways of the SOS response to evolving antibiotic resistance following a single treatment with ampicillin for 8 hr at 50 μg/mL. A mutant form of the SOS master regulator LexA (*lexA3*), which is incapable of being cleaved and thus defective in SOS induction, did not exhibit antibiotic resistance evolution (*Figure 3A*). Additionally, the deletion of either DpiB or DpiA (encoded by *citB* or *citA*, respectively), inhibiting the DpiBA two-component signal transduction system, did not result in the development of antibiotic resistance after ampicillin exposure (*Figure 3A*). Moreover, the deletion of several downstream effectors of the SOS response, including those involved in cell division inhibition (SulA and YmfM encoded by *sulA* and *ymfM*; *Simmons et al., 2008*; *Figure 3A*), and DNA repair (DNA Pol II, DNA Pol IV, and DNA Pol V encoded by *polB*, *dinB* and *umuDC*; *Ansari et al., 2015*) also did not lead to the evolution of antibiotic resistance (*Figure 3A*). These results indicate that the rapid resistance evolution observed in the *ΔrecA* strain is not mediated by SOS pathway mutants, suggesting that RecA governs resistance evolution through SOS-independent mechanisms.

In addition to the DNA repair components associated with the SOS response, DNA Pol I plays a role in processing RNA primers during lagging-strand synthesis and filling small gaps during DNA repair reactions (*Monk et al., 1971*). Since DNA Pol I (encoded by *polA*) has been demonstrated as an essential gene required for the growth of *E. coli* in rich medium, including the LB medium (*Joyce and Grindley, 1984*; *Konrad and Lehman, 1974*), we next utilised Single Molecule Localization Micros-copy (SMLM) to precisely locate the chromosome and DNA Pol I in a dynamic manner. During an 8 hr exposure to ampicillin, we observed the formation of multinucleated filaments in both the wild type and *ΔrecA* strain, indicating a pause in cell division and suggesting a time period for bacterial DNA repair to take place (*Figure 3B and C*; *Justice et al., 2008*). However, the expression level of DNA Pol I was significantly suppressed in the *ΔrecA* strain compared to the wild type strain after 4 hours of ampicillin exposure (*Figure 3D*). More notably, the super-resolution co-localisation analysis revealed a significantly lower ratio of co-localisation between the chromosome and DNA Pol I in the *ΔrecA* strain (*Figure 3E*). Together, these findings demonstrate that the RecA loss impairs DNA repair capacity beyond the SOS regulon. This repair deficiency contributes to genetic instability and is able to facili-tate the rapid evolution of antibiotic resistance through SOS-independent pathways.

## Repression of antioxidative gene expression promotes the evolution of antibiotic resistance in the *recA* mutant strain

To further comprehend the fast evolution of β-lactam resistance observed in this study, we inves-tigated the gene expression changes induced by ampicillin using a comprehensive transcriptome sequencing approach (total RNA-seq). Our analysis revealed significant transcriptomic alterations in

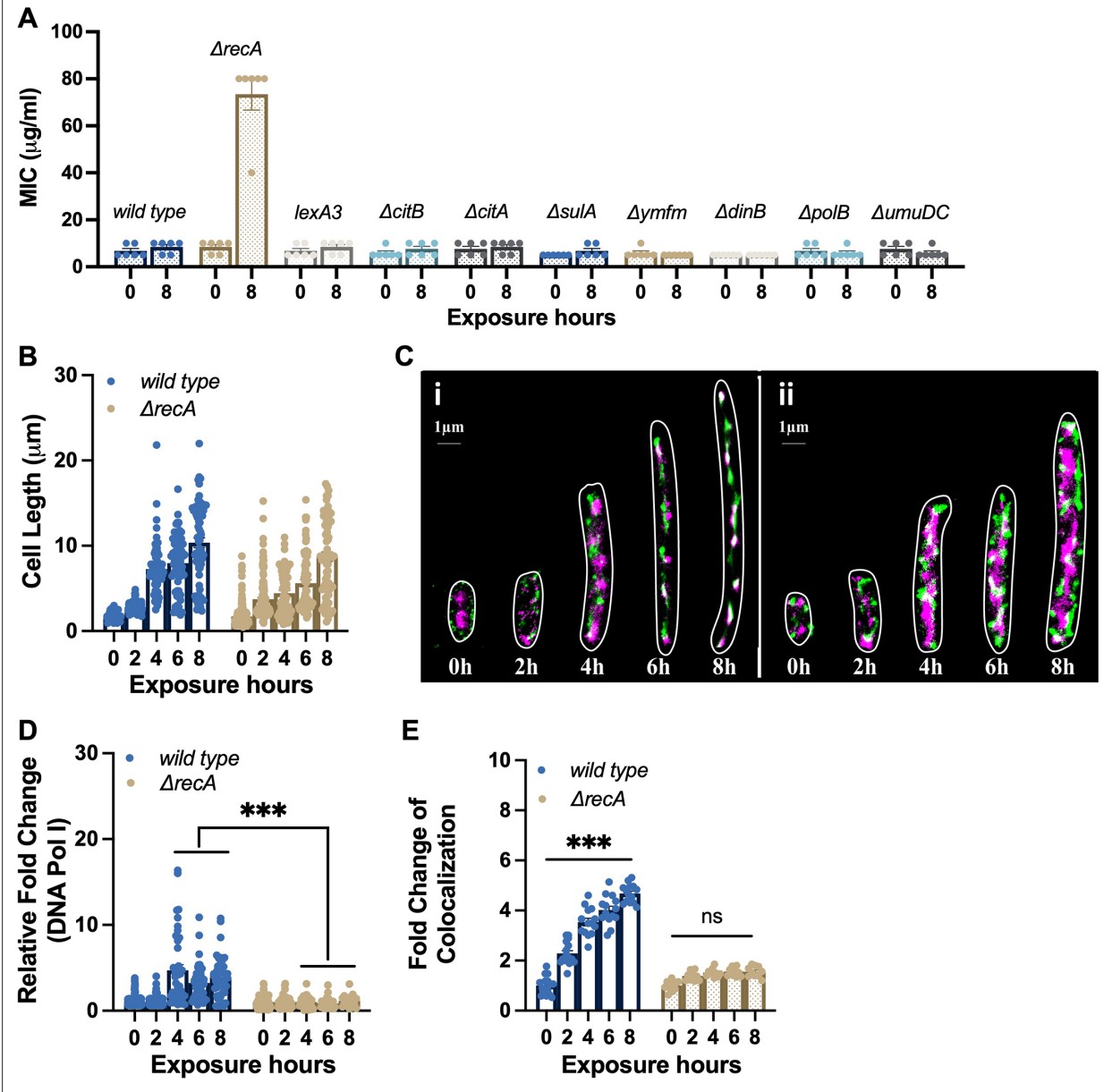

**Figure 3.** SOS-independent impairment of DNA repair in *ΔrecA*-resistant isolates. (**A**) MICs of ampicillin were tested against the wild type strain, *ΔrecA* strain, and mutants lacking specific genes from the SOS response after single exposures to ampicillin at 50 μg/mL for 8 hr. (**B**) Filament cell lengths in the wild type (n=*253*) and the *ΔrecA* strain (n=*216*) after single treatments with ampicillin at 50 μg/mL. (**C**) Multinucleated filaments were observed in the wild type (**i**) and the *ΔrecA* (**ii**) strain after single exposures to ampicillin at 50 μg/mL. Purple: *E. coli* chromosome; green: DNA Pol I. (**D**) Relative fold changes of DNA Pol I in the wild type and *ΔrecA* strain after single treatments with ampicillin at 50 μg/mL. (**E**) Co-localisation between the *E. coli* chromosome and DNA Pol I in the wild type and *ΔrecA* strain after the single exposures to ampicillin at 50 μg/mL. Data is shown as mean ± SEM. Significant differences among different treatment groups are analysed by independent t-test, *p<0.05, **p<0.01, ***p<0.001, *ns*, no significance.

both the wild type and *ΔrecA* strain isolates following a single treatment with ampicillin, compared to untreated controls (***Figure 4A***). Specifically, we identified changes in the expression of 161 and 248 coding sequences (with $log_2FC >2$ and p-value <0.05) in the wild type and *ΔrecA* strains, respectively. Principal component analysis (PCA) demonstrated a notable disparity in the effects of ampicillin on the *ΔrecA* strain compared to the wild type strain (***Figure 4B***). Additionally, Venn diagram analysis confirmed that 138 and 225 genes were uniquely regulated by ampicillin exposure in the wild type and *ΔrecA* strains, respectively (***Figure 4C***).

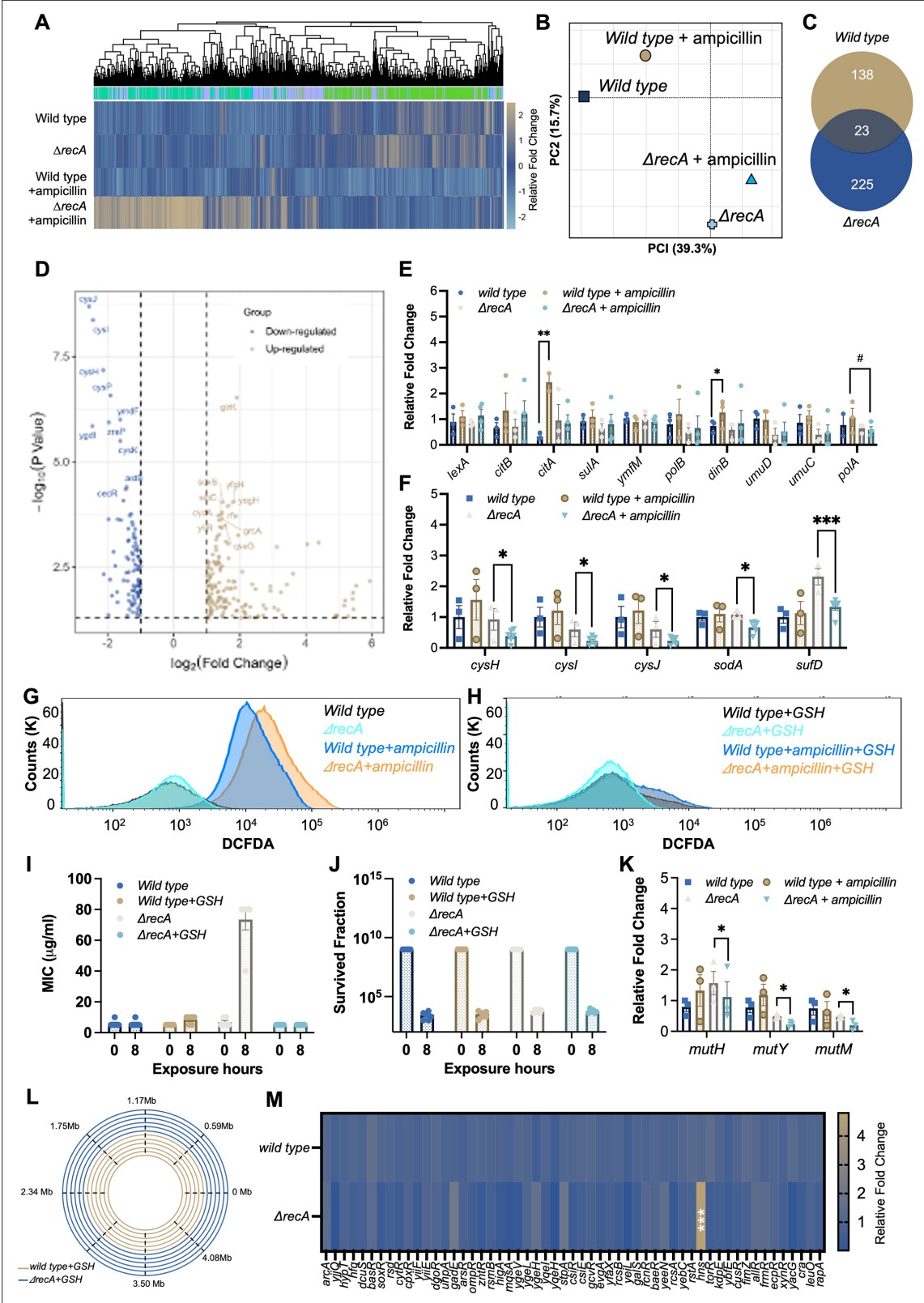

**Figure 4.** Overaccumulation of ROS drives the fast evolution of multi-drug resistance in the Δ*recA* strain. (**A**) Clustered heatmap of relative expression of coding sequences in the wild type and Δ*recA* strain with significant fold changes (log₂FC >2 and p-value <0.05). (**B**) Principal-component analysis (PCA) of normalised read counts for all strains. (**C**) Venn diagram of differentially expressed genes (log₂FC >2) after treatment with ampicillin at 50 µg/ml for 8 hours in the wild type and Δ*recA* strain. (**D**) The top 10 most differentially expressed genes in the Δ*recA* strain after the single treatment with

*Figure 4 continued on next page*

*Figure 4 continued*

ampicillin are labelled in each plot. Blue dots indicate genes with a significant downregulation compared to the untreated control ($\log_2$FC >2 and p-value <0.05), and yellow dots indicate genes with a significant upregulation compared to the untreated control ($\log_2$FC >2 and p-value <0.05). (**E**) Levels of transcription of SOS response system-associated genes and gene *polA* in the wild type and *ΔrecA* strain after single exposures to ampicillin for 8 hr. (**F**) Levels of transcription of different antioxidative associated genes in the wild type and *ΔrecA* strain after single exposures to ampicillin for 8 hr. (**G**) ROS levels were measured by flow cytometry before and after 8 hr of ampicillin treatment (50 µg/mL) in the wild type and *ΔrecA* strains. (**H**) ROS levels were measured by flow cytometry in the wild type and *ΔrecA* strains before and after 8 hr of ampicillin treatment at 50 µg/mL with the addition of GSH (50 mM). (**I**) The addition of 50 mM antioxidative compound GSH prevented the evolution of antibiotic resistance to ampicillin in the *ΔrecA* strain treated with ampicillin at 50 µg/mL for 8 hr. (**J**) Survival fraction after a single exposure to ampicillin at 50 µg/mL for 8 hr in the wild type and the *ΔrecA* strain with or without the addition of GSH at 50 mM. (**K**) Levels of transcription of proteins involved in the BER DNA repair system in the wild type and *ΔrecA* strain after single exposures to ampicillin for 8 hr. (**L**) Whole genome sequencing confirms undetectable DNA mutations in the wild type and *ΔrecA* strain treated with single exposures to ampicillin with the addition of GSH at 50 mM for 8 hr. (**M**) Transcription levels of all transcriptional repressors in the wild type and *ΔrecA* strain after single treatments with ampicillin for 8 hr. Total RNA-seq was performed with three repeats in each group. Each experiment was independently repeated at least six times, and the data are shown as mean ± SEM. Significant differences among different treatment groups are analysed by independent t-test, *p<0.05, **p<0.01, ***p<0.001, #p<0.05.

The online version of this article includes the following figure supplement(s) for figure 4:

**Figure supplement 1.** Transcriptional responses of the wild type and *ΔrecA* strain after single treatments with ampicillin for 8 hr.

To elucidate the differential expression of genes associated with specific biological functions, we conducted Gene Ontology (GO) enrichment analyses (*Figure 4—figure supplement 1A and B*) and Kyoto Encyclopedia of Genes and Genomes (KEGG) pathway analyses (*Figure 4—figure supplement 1C and D*). Our findings indicate that ampicillin profoundly impacted persistence pathways in the wild type strain, specifically affecting pathways related to quorum sensing, flagellar assembly, biofilm formation, and bacterial chemotaxis (*Helaine and Kugelberg, 2014*; *Spoering and Lewis, 2001*). Conversely, in the *ΔrecA* strain, a distinct functional category associated with the oxidative stress response exhibited significant and unique down-regulation. This category included activities such as sulfate transporter activity, iron-sulfur cluster assembly, oxidoreductase activity, and carboxylate reductase activity.

To identify specific genes showing significant fold changes ($\log_2$FC >2 and p value <0.05), we used volcano plots to visualise the comprehensive changes in gene expression across the genome (*Figure 4D*). We examined the transcription levels associated with the SOS response system and found that the transcription of several proteins in the wild type strain can be significantly induced by the single exposure to ampicillin, including *citB* and *dinB* (*Figure 4E*). However, in the *recA* mutant strain, antibiotic exposure does not affect the transcription levels of any SOS system-related proteins, suggesting that antibiotic exposure induced the SOS response in the wild type strain but not in the *ΔrecA* strain. More importantly, we discovered that the induction of the transcription level of DNA Pol I was significantly suppressed after the single treatment of ampicillin in the *ΔrecA* strain compared with that in the wild type strain (*Figure 4E*). This is consistent with our imaging results and further supports the notion that an SOS-independent evolutionary mechanism dominates the development of antibiotic resistance in *recA* mutant *E. coli*.

Further, significant downregulation in the transcription of antioxidative-related genes in the *ΔrecA* strain was detected, including *cysJ*, *cysI*, *cysH*, *soda*, and *sufD* (*Figure 4F*). This downregulation suggested an excessive accumulation of reactive oxygen species (ROS) due to compromised cell antioxidative defences. It has been previously reported that the induction of mutagenesis can be stimulated by the overproduction of ROS during antibiotic administration, leading to the evolution of antibiotic resistance both in vivo and in vitro (*Kohanski et al., 2010*; *Takahashi et al., 2017*). Therefore, we hypothesised that elevated ROS levels facilitate the rapid evolution of antibiotic resistance in the *ΔrecA* strain by increasing genetic instability and enabling the selection of resistant variants. To test it, we first examined the level of ROS generation in the wild type and *ΔrecA* strains treated with ampicillin for 8 hr by using the fluorescent probe DCFDA/H2DCFDA. We found that ROS levels significantly increased in both the wild type and *ΔrecA* strain after 8 hr of ampicillin treatment. However, ROS levels in the *ΔrecA* strain showed a significant further increase compared to the wild type strain (*Figure 4G*). Additionally, with the addition of 50 mM glutathione (GSH), a natural antioxidative compound, no significant change in ROS levels was observed in either the wild type or *ΔrecA* strain before and after ampicillin treatment (*Figure 4H*). Further, we supplemented the wild type and *ΔrecA* strains with 50 mM GSH and treated them with ampicillin at a concentration

of 50 µg/mL for 8 hr. Remarkably, the addition of GSH prevented the development of resistance to ampicillin in the Δ*recA* strain (**Figure 4I**), without impairing the bactericidal effectiveness of ampicillin (**Figure 4J**).

Apart from the SOS response, bacterial cells coordinate other DNA repair activities through a network of regulatory pathways, including base excision repair (BER) (**Foti et al., 2012**; **Friedberg et al., 2005**; **Hahm et al., 2022**). The excessive generation of ROS results in elevated levels of deoxy-8-oxo-guanosine triphosphate (8-oxo-dGTP), an oxidised form of dGTP that becomes both highly toxic and mutagenic upon integration into DNA. The presence of 8-oxo-dG can induce SNP mutations, especially those occurring in guanine, which can be actively rectified by the BER repair pathway (**Hahm et al., 2022**). Notably, BER glycosylases MutH and MutY can identify and repair these 8-oxo-dG-dependent mutations; however, when MutY and MutH are inactivated, unrepaired 8-oxo-dG can lead to the accumulation of SNP mutations within cells (**Hahm et al., 2022**). As a result, we conducted a further assessment of the transcription levels of the BER repair pathway in both the wild type and Δ*recA* strain before and after a single exposure to ampicillin. We discovered that after an 8 hr treatment of ampicillin, three DNA repair-associated proteins, including MutH, MutY, and MutM, were notably suppressed in the Δ*recA* strain (**Figure 4K**).

Finally, we sequenced the surviving Δ*recA* isolates and found that the addition of GSH inhibited drug resistance-associated mutations in the Δ*recA* strain, which were detected in the Δ*recA*-resistant isolates including genes of the promoter of *ampC* and *acrB* (**Figure 4L**). Given the DNA repair impairment resulting in the generation of ROS, it would have been expected for genes involved in the oxidative stress response to be induced in RecA-deficient cells. However, the repression of antioxidative-related genes indicated the involvement of transcriptional repressors that might be regulated by RecA. Consequently, we examined the transcription levels of all transcriptional repressors in both the wild type and Δ*recA* strain. Remarkably, we observed a significant upregulation of H-NS, a crucial transcriptional repressor (**Figure 4M**). This finding suggests that the upregulation of H-NS could potentially contribute to the suppression of antioxidant gene expression in the Δ*recA* strain, thereby promoting ROS accumulation and subsequent resistance evolution. Together, these findings demonstrate that RecA deficiency not only impairs DNA repair but also suppresses the oxidative stress response, leading to elevated ROS and increased mutational load. This oxidative-genetic imbalance forms the basis for enhanced mutational supply in RecA-deficient cells.

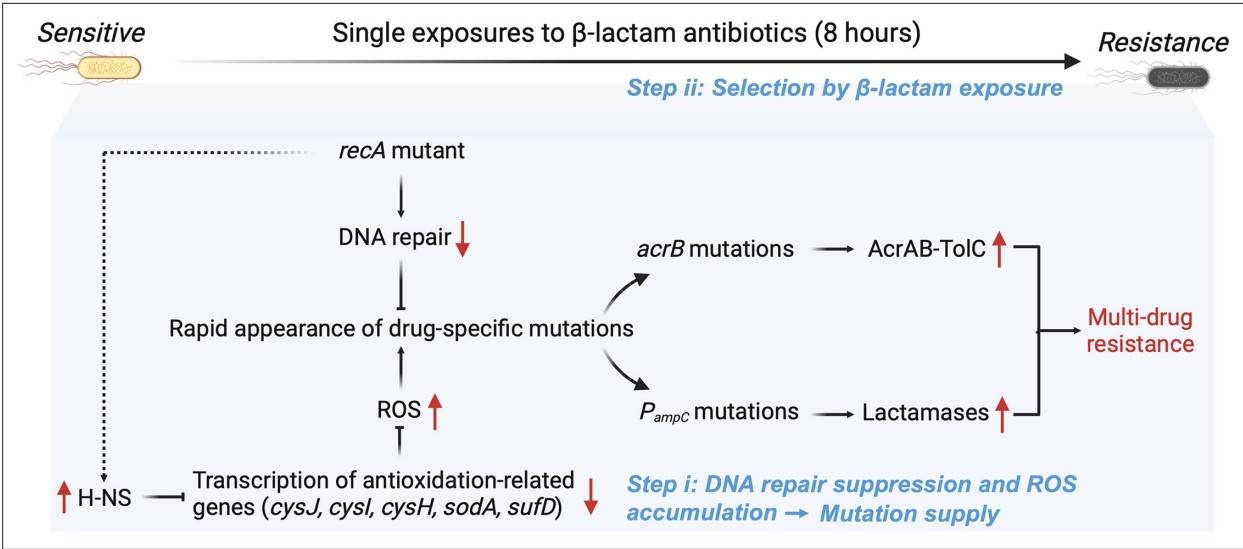

**Figure 5.** Mechanism of rapid development of multidrug resistance in the *recA* mutant *E. coli* strain. Deletion of *recA* impairs DNA damage repair and upregulates the transcription of the global repressor H-NS, though the mechanism of this regulation remains unclear. Elevated H-NS levels repress the expression of multiple antioxidant-related genes (*cysI, cysJ, cysH, sodA, sufD*), leading to excessive accumulation of ROS. The resulting oxidative stress increases the overall mutational burden. Upon single exposure to β-lactam antibiotics, drug-resistant subpopulations carrying specific mutations, such as in *acrB* or $P_{ampC}$, are selectively enriched, ultimately driving the rapid emergence of multidrug resistance. This process reflects a two-step mechanism involving enhanced mutational supply and subsequent selection under antibiotic pressure.

## Discussion

In this study, we challenge the prevailing view that disabling RecA and thereby inhibiting the SOS response can prevent bacteria from developing antibiotic resistance (*Pavlopoulou, 2018*). While the SOS response and RecA have been extensively studied for their roles in antibiotic resistance evolution (*Pavlopoulou, 2018*), we observed a remarkably fast and stable evolution of multidrug resistance in the *E. coli* Δ*recA* strains following a single exposure to β-lactam antibiotics. This phenomenon cannot be explained by canonical SOS-driven mutagenesis (*Janion, 2008*), but instead reflects the interplay of two distinct evolutionary forces: RecA loss increases mutational supply through DNA repair suppression and ROS accumulation, while antibiotic-induced lethal stress provides a selective environment that promotes the expansion of rare and resistance-conferring variants (*Figure 5*).

To determine whether antibiotic exposure in Δ*recA* cells directly induces mutations or selectively enriches resistant variants, we combined statistical modelling, fluctuation analysis, and whole-genome sequencing. Although mutation rates estimated by maximum likelihood were moderately elevated in Δ*recA* cells after ampicillin treatment, the mutation frequency distributions were highly right-skewed and deviated from Poisson expectations, a hallmark of clonal selection rather than uniform and population-wide mutagenesis. These findings align with the classical Luria-Delbrück framework and indicate that resistance evolution in Δ*recA* is primarily driven by selection (*Luria and Delbrück, 1943*). However, our data also demonstrate that the mutational supply is enhanced in this background due to antibiotic-induced oxidative stress and impaired DNA repair, which together increase the likelihood that resistance-conferring mutations arise and persist. Thus, this process reflects a selection-dominated evolutionary trajectory facilitated by stress-enhanced mutagenesis, rather than classical induced mutation or selection on pre-existing variants.

Mechanistically, our data reveal that RecA plays broader roles in genome stability beyond its function in SOS activation. Disruption of individual SOS components, including *lexA*, *citA/B*, or translesion polymerases, did not recapitulate the Δ*recA*-specific resistance phenotype. Instead, Δ*recA* strains exhibited suppression of DNA Pol I and key BER genes, such as *mutH*, *mutY*, and *mutM*, confirming that RecA is essential for the maintenance of repair fidelity under stress conditions. This defect likely enables ROS-induced DNA lesions to persist and become fixed as mutations.

In parallel, we identified a striking repression of antioxidative stress response genes in Δ*recA* strains exposed to ampicillin, including *cysJIH*, *sodA*, and *sufD*. This transcriptional suppression was associated with markedly elevated ROS levels. Crucially, supplementation with the antioxidant GSH reversed both ROS accumulation and the emergence of resistance, without impairing the bactericidal activity of ampicillin. This decouples the mutagenic and lethal effects of ROS and highlights ROS as a driver of mutational supply rather than survival.

To explore the transcriptional basis for oxidative dysregulation, we examined global repressors and discovered that *hns* was significantly upregulated in Δ*recA* strains after ampicillin exposure. H-NS is a known transcriptional silencer of stress response genes, including those involved in redox regulation (*Dorman, 2004*; *Bracco et al., 1989*). We propose that RecA deficiency may lead to H-NS derepression, thereby silencing antioxidant defences, exacerbating ROS accumulation and enabling mutation accumulation under stress. Although previous findings report that H-NS can down-regulate the transcriptional expression of *cysIJH* through the mediation of *cysB* (*Álvarez et al., 2015*; *Ishihama and Shimada, 2021*), the mechanistic relationship between RecA and H-NS regulation remains to be experimentally validated.

More broadly, our findings highlight the repair-redox axis as a central regulator of bacterial evolvability. Rather than solely targeting bacterial growth or survival, future antimicrobial strategies might focus on constraining mutational potential. For example, co-administration of antioxidants or repair stabilisers could buffer stress-induced mutagenesis without compromising antibiotic efficacy, which has been a concept already under exploration in cancer therapeutics (*Aguilera and Gómez-González, 2008*; *Klinakis et al., 2020*).

Clinically, the emergence of *acrB* mutations and enhanced activity of the AcrAB-TolC efflux pump in Δ*recA* strains recapitulates known multidrug resistance pathways. These observations raise concerns for therapies combining DNA repair inhibitors with ROS-inducing antibiotics or anticancer drugs (*Wang et al., 2021*). Our data suggest that such combinations may inadvertently promote resistance evolution, particularly in immunocompromised patients or during chemotherapy.

Finally, ROS-driven mutagenesis and repair suppression are not unique to β-lactams. Antimicrobial technologies such as antimicrobial photodynamic therapy (aPDT) and cold atmospheric plasma (CAP) (*Wilson and Patterson, 2008*; *Vatansever et al., 2013*) also rely on oxidative stress. In RecA-deficient or stress-sensitised bacteria, these approaches may risk accelerating resistance evolution unless accompanied by safeguards that preserve genomic integrity. Thus, maintaining the transcription of genes involved in the oxidative stress defence or combining antibiotics represents a promising strategy to prevent ROS-driven mutagenesis and thereby limit the evolutionary emergence of resistance during antimicrobial therapy.

## Materials and methods

### Bacterial strains, medium, and antibiotics

Bacterial strains and plasmids used in this work are described in *Supplementary file 2*, *Supplementary file 3*. Luria-Bertani (LB) was used as broth or in agar plates. *E. coli* cells were grown in LB liquid medium or on LB agar (1.5% w/v) plates at 37 °C. Unless stated otherwise, antibiotics were supplemented, where appropriate. Antibiotic stock solutions were prepared by dissolving antibiotics in MilliQ filter sterilising, including ampicillin (50 mg/mL), penicillin G (100 mg/mL), carbenicillin (20 mg/mL), kanamycin (50 mg/mL) and tetracycline (10 mg/mL). Chloramphenicol stock solution was prepared in 95% EtOH (25 mg/mL). Antibiotic solutions were stored at –20 °C (long-term) or 4 °C (short-term).

### Treatment with antibiotics to induce evolutionary resistance

For the single exposure to antibiotic experiment, an overnight culture (0.6 mL; $1 \times 10^9$ CFU/mL cells) was diluted 1:50 into 30 mL LB medium supplemented with antibiotics (50 µg/mL ampicillin, 1 mg/mL penicillin G, or 200 µg/mL carbenicillin) and incubated at 37 °C with shaking at 250 rpm for 0, 2, 4, 6, and 8 hrs, respectively. After each treatment, the antibiotic-containing medium was removed by washing twice (20 min centrifugation at 1500 × *g*) in fresh LB medium (See *Figure 1A* for method overview).

To test resistance, the surviving isolates were first resuspended in 30 mL LB medium and grown overnight at 37 °C with shaking at 250 rpm. The regrown culture was then plated onto LB agar and incubated overnight at 37 °C. Single colonies were isolated and grown in LB medium for 4–6 hours at 37 °C with shaking at 250 rpm, which were then used to test the resistance or stored at –80 °C for future use.

For the ALE antibiotic treatment experiments, an overnight culture (0.6 mL; $1 \times 10^9$ CFU/mL cells) was diluted 1:50 into 30 mL LB medium supplemented with 50 µg/mL ampicillin and incubated at 37 °C with shaking at 250 rpm for 4.5 hr. After treatment, the antibiotic-containing medium was removed by washing twice (20 min centrifugation at 1500 × *g*) in a fresh LB medium. The remaining pellet was resuspended in 30 mL LB medium and grown overnight at 37 °C with shaking at 250 rpm. Ampicillin treatment was applied to the regrown culture and repeated until resistance was established, as confirmed by MIC measurement.

### Antibiotic susceptibility testing

The susceptibility of *E. coli* cells to antibiotics was measured using minimum inhibitory concentration (MIC) testing (*Scholar and Pratt, 2000*). In brief, overnight cultures were diluted and incubated at 37 °C for 4–6 hr with shaking at 250 rpm. Cells were then diluted 1:100 and incubated with increasing concentrations of antibiotics in the Synergy HT BioTek plate reader (BioTek Instruments Inc, USA) at 37 °C for 16 hr. It was programmed to measure the OD hourly at 595 nm (Gen5 software, BioTek Instruments Inc, USA). The minimum inhibitory concentration was determined as the concentration of antibiotic where no visible growth was observed.

### Survival assays

Overnight cultures of *E. coli* were prepared from single colonies in LB medium and incubated at 37 °C with shaking (250 rpm). The overnight cultures were diluted 1:50 in fresh LB medium containing 50 µg/mL ampicillin to initiate antibiotic treatment. Cultures were incubated at 37 °C for the indicated times under shaking conditions. Following treatment, cells were collected by centrifugation (1500 g, 20 minutes), and the antibiotic-containing medium was removed by washing twice with fresh

LB medium. Serial dilutions of the washed cultures were prepared, and 25 µL of each dilution was plated onto LB agar plates. Plates were incubated overnight at 37 °C, and CFUs were counted the following day to evaluate bacterial survival rates.

## Mutation frequency and fluctuation analysis

Overnight cultures inoculated from single colonies in LB medium were diluted 1:1,000,000 and incubated at 37 °C with shaking until the $OD_{600}$ reached 1~1.3. This extreme dilution minimises the presence of pre-existing stationary phase mutants and allows de novo mutation events to occur during exponential growth. For each biological condition, 96 independent parallel cultures were prepared to perform a fluctuation analysis. The total number of colony-forming units per ml (CFU/ml) was determined by plating on LB agar. To identify rifampicin-resistant mutants, the remaining culture volume was centrifuged and plated on LB agar containing rifampicin (100 µg/mL). LB plates were incubated for 24 hr at 37 °C and selective plates were incubated for 48–72 hr at 37 °C (*Gutierrez et al., 2013*).

The mutation frequency was calculated as the ratio of CFU/ml on rifampicin-containing plates to CFU/ml on non-selective LB plates for each culture. Distributions of mutation frequencies across replicate cultures were plotted, and deviations from Poisson expectations were assessed by model fitting.

To estimate the mutation rate (mutations per culture), maximum likelihood estimation (MLE) was applied using the Ma-Sandri-Sarkar algorithm implemented in the FALCOR toolset (https://www.keshavsingh.org/protocols/FALCOR.html). The inferred mutation rate distributions were compared across treatment conditions, and non-Poisson distributions indicative of jackpot cultures were further analysed using fluctuation test-based inference following the Luria–Delbrück framework (*Luria and Delbrück, 1943*).

## Construction of *recA* deletion mutant

Lambda Red recombination was used to generate the gene *recA* deletion in the *E. coli* K-12 strain, followed by previously reported methods with modifications (*Datsenko and Wanner, 2000*; *Baba et al., 2006*). Primers (*recA-FWD* and *recA-REV, Supplementary file 4*) were designed approximately 50 bp upstream and downstream to the gene *recA* on the chromosome to amplify the tetracycline cassette as well as the flanking DNA sequence needed for homologous recombination. Phusion polymerase (NEB) was used to amplify the DNA sequence (*Supplementary file 4*), and the reaction was cleaned up using a PureLink PCR purification kit (Thermo Fisher Scientific) as per the manufacturer's instructions. Electro-competent *E. coli* MG1655 containing the recombinase plasmid pKD46 was transformed with 50 ng of amplified DNA at 30 °C. The transformation was plated onto LB agar plates containing 10 µg/mL tetracycline and incubated overnight at 37 °C. PCR was used to confirm the insertion of the tetracycline resistance cassette at the correct site on the chromosome using primers upstream and downstream to the gene *recA*. The newly constructed mutant strains were cured of plasmid pKD46 by incubating LB streak plates at 42 °C overnight. Loss of the plasmid was confirmed by lack of ampicillin sensitivity on LB agar plates. Mutant strains were made electro-competent, and 50 µL of cells were transformed with plasmid pCP20 and incubated on 100 µg/mL ampicillin plates at 30 °C overnight. A few colonies were then restreaked onto LB plates and incubated overnight at 42 °C. PCR products confirmed the loss of cassette and plasmid.

## β-lactamase assay

The amount of β-lactamase was measured using a β-lactamase Activity Assay Kit (Sigma-Aldrich, US). Briefly, cells were collected by centrifugation at 10,000 × *g* for 10 min, and the pellet was resuspended with 5 µL of assay buffer per mg of sample. Then, 48 µL of the sample was mixed with 2 µL of nitrocefin. The β-Lactamase activity was monitored by measuring the absorbance at 490 nm for 30 min at 28 °C. The level of β-lactamase was determined by the absorbance at $OD_{390}$.

## Whole genome sequencing

Resistant clones were isolated by selection using LB agar plates with the supplementation of ampicillin at 50 µg/mL, and non-resistant clones were isolated from the LB agar plates without the supplementation of antibiotics. Chromosomal DNA was extracted and purified using the PureLink Genomic DNA mini kit following the manufacturer's instructions (Thermo Fisher Scientific). Whole genome sequencing (WGS) was conducted following the Nextera Flex library preparation kit process

(Illumina). Briefly, genomic DNA was quantitatively assessed using the Quant-iT picogreen dsDNA assay kit (Invitrogen, USA). The sample was normalised to the concentration of 1 ng/µL. 10 ng of DNA was used for library preparation. After tagmentation, the tagmented DNA was amplified using the facility's custom-designed i7 or i5 barcodes, with 12 cycles of PCR. The quality control for the samples was done by sequencing a pool of samples using MiSeq V2 nano kit - 300 cycles. After library amplification, 3 µL of each library was pooled into a library pool. The pool was then cleaned up using SPRI beads following the Nextera Flex clean-up and size selection protocol. The pool was then sequenced using a MiSeq V2 nano kit (Illumina, USA). Based on the sequencing data generated, the read count for each sample was used to identify the failed libraries (i.e. libraries with less than 100 reads).

Moreover, libraries were pooled at different amounts based on the read count to ensure equal representation in the final pool. The final pool was sequenced on Illumina NovaSeq 6000 Xp S4 lane, 2×150 bp. WGS read quality was assessed using FASTQC (version 0.11.5) and trimmed using Trimmomatic (version 0.36) with default parameters and trimmed of adaptor sequences (TruSeq3 paired-ended). Reads were aligned to the *E. coli* MG1655 genome (http://bacteria.ensembl.org/Escherichia_coli_str_k_12_substr_mg1655_gca_000005845/Info/Index/, assembly ASM584v2) and then analysed variants following GATK Best Practices for Variant Discovery (HaplotypeCaller) (*Van der Auwera and O'Connor, 2020*). Further genome variant annotation was conducted using the software SnpEff (*Cingolani et al., 2012*).

## Global transcriptome sequencing

After ampicillin treatment for 0 and 8, surviving isolates were immediately washed and harvested for global transcriptome sequencing. Total RNA was extracted from the cell pellets using a Pure-Link RNA mini kit (Invitrogen) as per the manufacturer's instructions. The global transcriptome sequencing was processed and analysed by Genewiz, Jiangsu, China. Primers used in this work are listed in *Supplementary file 4*. RNA-Seq read quality was assessed using FASTQC and trimmed using Trimmomatic with default parameters. Reads were aligned to the *E. coli* MG1655 genome (http://bacteria.ensembl.org/Escherichia_coli_str_k_12_substr_mg1655_gca_000005845/Info/Index/, assembly ASM584v2) and then counted using the RSubread aligner with default parameters (*Liao et al., 2019*). After mapping, differential expression analysis was carried out using strand-specific gene-wise quantification using the DESeq2 package (*Love et al., 2014*). Further normalisation was conducted using RUVSeq and the RUV correction method, with $k = 1$ to correct for batch effects, using replicate samples to estimate the factors of unwanted variation (*Risso et al., 2014*). Absolute counts were transformed into standard z-scores for each gene over all treatments, that is, absolute read for a gene minus mean read count for that gene over all samples and then divided by the standard deviation for all counts over all samples. Genes with an adjusted *P* value ($P_{adj}$) of ≤0.05 were considered differentially expressed. PseudoCAP analysis was conducted by calculating the percentage of genes in each classification that were differentially expressed ($log_2FC \geq \pm 2$, $P_{adj} \leq 0.05$).

## ROS measurement

Intracellular ROS accumulation was measured using the Cellular ROS Assay Kit (Abcam, ab113851) according to the manufacturer's protocol. Overnight cultures of both wild type and Δ*recA* strains were grown in LB medium. Cells were treated with 50 µg/mL ampicillin for 8 hours at 37 °C with shaking (250 rpm). After the treatment, cells were washed twice with 1 x buffer to remove residual antibiotics and debris. The collected cells were resuspended in 1 x buffer and incubated with the fluorescent ROS probe DCFDA (2',7'–dichlorofluorescin diacetate)/H2DCFDA at a final concentration of 10 µM for 30 min at 37 °C in the dark to prevent probe degradation.

After incubation, 500 µL of the stained cells were immediately analysed using a CytoFLEX LX flow cytometer (Beckman Coulter). Fluorescence was detected in the FITC channel (excitation at 488 nm, emission at 525 nm). Data were analysed using FlowJo software (Version X, Tree Star Inc), with fluorescence intensity serving as an indicator of intracellular ROS levels. Appropriate controls, including unstained cells and cells without ampicillin treatment, were included to ensure accurate ROS measurement.

## Single-molecule localisation imaging and data analysis

Single-molecule localisation imaging was performed on a custom-built Stochastic Optical Reconstruction Microscope (STORM) with an Olympus IX81 microscope frame, a ×100 magnification NA 1.45 objective (Olympus) and an EMCCD camera (DU-897, Andor) as described previously (*Su et al., 2020*; *Wang et al., 2015*; *Huang et al., 2008*). In summary, 35 mm cell culture dishes (0.17 mm No.1 coverglass) were cleaned with 1 M KOH for 30 min in an ultrasonic cleaning machine, followed by three washes with MilliQ water. The dishes were air-dried with high-purity nitrogen blowing and sterilised by UV exposure for 30 min. *E. coli* cells were fixed with NaPO4 (30 nM), formaldehyde (2.4%), and glutaraldehyde (0.04%) at room temperature for 15 min, followed by 45 min on ice. Samples were then centrifuged to collect the pellet cells, and the supernatant was discarded. Cell pellets were washed twice with phosphate-buffered saline (PBS), pH 7.4. Cells were resuspended in 200 µL of GTE buffer and kept on ice until 200 µL was placed onto the coverslip bottom of the cleaned 35 mm culture dish. To label the bacterial chromosome, a Click-iT EdU kit was used prior to fixation following the manufacturer's instruction (ThermoFisher) and as described before. To label DNA polymerase I, fixed cells were blocked and permeabilised with blocking buffer (5% wt/vol bovine serum albumin (Sigma-Aldrich) and 0.5% vol/vol Triton X-100 in PBS) for 30 min and then incubated with 1 µg/mL primary antibody against DNA polymerase I (ab188424, Abcam) in blocking buffer for 60 min at room temperature. After washing with PBS three times, the cells were incubated with 2 µg/mL fluorescently labelled secondary antibody (Alexa 647, A20006, Thermo Fisher) against the primary antibody in the blocking buffer for 40 min at room temperature. After washing with PBS three times, the cells were postfixed with 4% (wt/vol) paraformaldehyde in PBS for 10 min and stored in PBS before imaging. STORM image analysis, drift correction, image rendering, protein cluster identification and images presentation were performed using Insight342, custom-written Matlab (2012a, MathWorks) codes, SR-Tesseler (IINS, Interdisciplinary Institute for Neuroscience; *Levet et al., 2015*), and Image J (National Institutes of Health).

## Statistical analysis

Statistical analysis was performed using GraphPad Prism v.9.0.0. All data are presented as individual values and mean or mean ± SEM. Unless otherwise specified, statistical comparisons were conducted using one-way ANOVA (for multiple groups) or unpaired two-tailed Student's t-tests (for two-group comparisons), assuming a 95% confidence interval. A probability value of $p < 0.05$ was considered significant. Statistical significance is indicated in each figure. All remaining experiments were repeated independently, at least six with similar results.

## Acknowledgements

This work was supported by the Australian Research Council (ARC grant no: APP1165135), Science and Technology Innovation Commission of Shenzhen (KQTD20170810110913065), Australia China Science and Research Fund Joint Research Centre for Point-of-Care Testing (ACSRF658277, SQ2017YFGH001190).

## Additional information

### Funding

| Funder | Grant reference number | Author |
| --- | --- | --- |
| Australian Research Council | APP1165135 | Dayong Jin |
| Australian Research Council | ACSRF658277 | Dayong Jin |
| Science and technology innovation commission of shen zhen | KQTD20170810110913065 | Dayong Jin |

| Funder | Grant reference number | Author |
|---|---|---|
| Australia China Science and Research | SQ2017YFGH001190 | Dayong Jin Qian Peter Su |
| National Health and Medical Research Council | APP1177374 | Qian Peter Su |

The funders had no role in study design, data collection and interpretation, or the decision to submit the work for publication.

## Author contributions

Le Zhang, Conceptualization, Data curation, Formal analysis, Validation, Investigation, Methodology, Writing – original draft, Project administration, Writing – review and editing; Yunpeng Guan, Data curation, Formal analysis, Validation, Investigation, Methodology; YuenYee Cheng, Data curation, Formal analysis, Supervision, Writing – review and editing; Nural N Cokcetin, Amy L Bottomley, Formal analysis, Supervision, Writing – original draft, Writing – review and editing; Andrew Robinson, Formal analysis, Methodology; Elizabeth J Harry, Antoine M van Oijen, Formal analysis, Supervision, Funding acquisition, Writing – original draft, Writing – review and editing; Qian Peter Su, Conceptualization, Data curation, Software, Formal analysis, Supervision, Investigation, Methodology, Writing – original draft; Dayong Jin, Conceptualization, Supervision, Funding acquisition, Writing – original draft, Writing – review and editing

## Author ORCIDs

Le Zhang https://orcid.org/0000-0003-0641-2383
Andrew Robinson https://orcid.org/0000-0002-3544-0976
Antoine M van Oijen https://orcid.org/0000-0002-1794-5161
Qian Peter Su https://orcid.org/0000-0001-7364-3945
Dayong Jin https://orcid.org/0000-0003-1046-2666

Reviewer #1 (Public review): https://doi.org/10.7554/eLife.95058.4.sa1
Reviewer #3 (Public review): https://doi.org/10.7554/eLife.95058.4.sa2
Author response https://doi.org/10.7554/eLife.95058.4.sa3

# Additional files

## Supplementary files

Supplementary file 1. Other mutations detected in the ΔrecA resistant isolates. Other mutations detected in the ΔrecA resistant isolates.

Supplementary file 2. Strains used in this study. Strains used in this study.

Supplementary file 3. Plasmids used in this study. Plasmids used in this study (*Ghodke et al., 2019*).

Supplementary file 4. Primers used in this study. Primers used in this study.

MDAR checklist

Source code 1. STORM Code.

## Data availability

Sequence data supporting this study's findings have been deposited in the GEO repository with the GEO accession number GSE179434.

The following dataset was generated:

| Author(s) | Year | Dataset title | Dataset URL | Database and Identifier |
|---|---|---|---|---|
| Guan YP, Zhang L | 2021 | Next Generation Sequencing Facilitates Quantitative Analysis of *E. coli*, Wild Type and ΔrecA Transcriptomes | https://www.ncbi.nlm.nih.gov/geo/query/acc.cgi?acc=GSE179434 | NCBI Gene Expression Omnibus, GSE179434 |

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
