## [Editor Report · eLife Assessment]

This study presents a **valuable** observation of how deletion of a major repair protein in bacteria can facilitate the rise of mutations that confer resistance against a range of different antibiotics. The data presented are **convincing**, and the authors addressed the concerns raised by the reviewers in their resubmission, improving the strength of their findings.

---

## [Referee Report · Reviewer #1 (Public review)]

Summary:

Jin et al. investigated how the bacterial DNA damage (SOS) response and its regulator protein RecA affects the development of drug resistance under short-term exposure to beta-lactam antibiotics. Canonically, the SOS response is triggered by DNA damage, which results in the induction of error-prone DNA repair mechanisms. These error-prone repair pathways can increase mutagenesis in the cell, leading to the evolution of drug resistance. Thus, inhibiting the SOS regulator RecA has been proposed as means to delay the rise of resistance.

In this paper, the authors deleted the RecA protein from *E. coli* and exposed this ∆recA strain to selective levels of the beta-lactam antibiotic, ampicillin. After an 8h treatment, they washed the antibiotic away and allowed the surviving cells to recover in regular media. They then measured the minimum inhibitory concentration (MIC) of ampicillin against these treated strains. They note that after just 8 h treatment with ampicillin, the ∆recA had developed higher MICs towards ampicillin, while by contrast, wild-type cells exhibited unchanged MICs. This MIC increase was also observed in subsequent generations of bacteria, suggesting that the phenotype is driven by a genetic change.

The authors then used whole genome sequencing (WGS) to identify mutations that accounted for the resistance phenotype. Within resistant populations, they discovered key mutations in the promoter region of the beta-lactamase gene, ampC; in the penicillin-binding protein PBP3 which is the target of ampicillin; and in the AcrB subunit of the AcrAB-TolC efflux machinery. Importantly, mutations in the efflux machinery can impact the resistance towards other antibiotics, not just beta-lactams. To test this, they repeated the MIC experiments with other classes of antibiotics, including kanamycin, chloramphenicol, and rifampicin. Interestingly, they observed that the ∆recA strains pre-treated with ampicillin showed higher MICs towards all other antibiotics tested. This suggests that the mutations conferring resistance to ampicillin are also increasing resistance to other antibiotics.

The authors then performed an impressive series of genetic, microscopy, and transcriptomic experiments to show that this increase in resistance is not driven by the SOS response, but by independent DNA repair and stress response pathways. Specifically, they show that deletion of the recA reduces the bacterium's ability to process reactive oxygen species (ROS) and repair its DNA. These factors drive the accumulation of mutations that can confer resistance towards different classes of antibiotics. The conclusions are reasonably well-supported by the data, but some aspects of the data and the model need to be clarified and extended.

Strengths:

A major strength of the paper is the detailed bacterial genetics and transcriptomics that the authors performed to elucidate the molecular pathways responsible for this increased resistance. They systemically deleted or inactivated genes involved in the SOS response in *E. coli*. They then subjected these mutants to the same MIC assays as described previously. Surprisingly, none of the other SOS gene deletions resulted in an increase in drug resistance, suggesting that the SOS response is not involved in this phenotype. This led the authors to focus on the localization of DNA PolI, which also participates in DNA damage repair. Using microscopy, they discovered that in the RecA deletion background, PolI co-localizes with the bacterial chromosome at much lower rates than wild-type. This led the authors to conclude that deletion of RecA hinders PolI and DNA repair. Although the authors do not provide a mechanism, this observation is nonetheless valuable for the field and can stimulate further investigations in the future.

In order to understand how RecA deletion affects cellular physiology, the authors performed RNA-seq on ampicillin-treated strains. Crucially, they discovered that in the RecA deletion strain, genes associated with antioxidative activity (cysJ, cysI, cysH, soda, sufD) and Base Excision Repair repair (mutH, mutY, mutM), which repairs oxidized forms of guanine, were all downregulated. The authors conclude that down-regulation of these genes might result in elevated levels of reactive oxygen species in the cells, which in turn, might drive the rise of resistance. Experimentally, they further demonstrated that treating the ∆recA strain with an antioxidant GSH prevents the rise of MICs. These observations will be useful for more detailed mechanistic follow-ups in the future.

Weaknesses:

Throughout the paper, the authors use language suggesting that ampicillin treatment of the ∆recA strain induces higher levels of mutagenesis inside the cells, leading to the rapid rise of resistance mutations. However, as the authors note, the mutants enriched by ampicillin selection can play a role in efflux and can thus change a bacterium's sensitivity to a wide range of antibiotics, in what is known as cross-resistance. The current data is not clear on whether the elevated "mutagenesis" is driven by ampicillin selection or by a bona fide increase in mutation rate.

Furthermore, on a technical level, the authors employed WGS to identify resistance mutations in the ampicillin-treated wild-type and ∆recA strains. However, the WGS methodology described in the paper is inconsistent. Notably, wild-type WGS samples were picked from non-selective plates, while ΔrecA WGS isolates were picked from selective plates with 50 μg/mL ampicillin. Such an approach biases the frequency and identity of the mutations seen in the WGS and cannot be used to support the idea that ampicillin treatment induces higher levels of mutagenesis.

Finally, it is important to establish what the basal mutation rates of both the WT and ∆recA strains are. Currently, only the ampicillin-treated populations were reported. It is possible that the ∆recA strain has inherently higher mutagenesis than WT, with a larger subpopulation of resistant clones. Thus, ampicillin treatment might not, in fact, induce higher mutagenesis in ∆recA.

Summary of revised manuscript:

In their revisions, the authors have addressed my major concerns with additional experiments and changes to the text. Thank you!

---

## [Referee Report · Reviewer #3 (Public review)]

Summary:

In the present work, Zhang et al investigate the involvement of the bacterial DNA damage repair SOS response in the evolution of beta-lactam drug resistance in *Escherichia coli*. Using a combination of microbiological, bacterial genetics, laboratory evolution, next-generation, and live-cell imaging approaches, the authors propose short-term (transient) drug resistance evolution can take place in RecA-deficient cells in an SOS response-independent manner. They propose the evolvability of drug resistance is alternatively driven by the oxidative stress imposed by accumulation of reactive oxygen species and compromised DNA repair. Overall, this is a nice study that addresses a growing and fundamental global health challenge (antimicrobial resistance).

Strengths:

The authors introduce new concepts to antimicrobial resistance evolution mechanisms. They show short-term exposure to beta-lactams can induce durably fixed antimicrobial resistance mutations. They propose this is due to compromised DNA repair and oxidative stress. Antibiotic resistance evolution under transient stress is poorly studied, so the authors' work is a nice mechanistic contribution to this field.

Weaknesses:

The authors revisions have significantly addressed weaknesses previously identified earlier in the review process.

---

## [Author Response]

The following is the authors’ response to the previous reviews

**Reviewer #1 (Public review):**
Jin et al. investigated how the bacterial DNA damage (SOS) response and its regulator protein RecA affects the development of drug resistance under short-term exposure to beta-lactam antibiotics. Canonically, the SOS response is triggered by DNA damage, which results in the induction of error-prone DNA repair mechanisms. These error-prone repair pathways can increase mutagenesis in the cell, leading to the evolution of drug resistance. Thus, inhibiting the SOS regulator RecA has been proposed as means to delay the rise of resistance.In this paper, the authors deleted the RecA protein from *E. coli* and exposed this ∆recA strain to selective levels of the beta-lactam antibiotic, ampicillin. After an 8h treatment, they washed the antibiotic away and allowed the surviving cells to recover in regular media. They then measured the minimum inhibitory concentration (MIC) of ampicillin against these treated strains. They note that after just 8 h treatment with ampicillin, the ∆recA had developed higher MICs towards ampicillin, while by contrast, wild-type cells exhibited unchanged MICs. This MIC increase was also observed subsequent generations of bacteria, suggesting that the phenotype is driven by a genetic change.The authors then used whole genome sequencing (WGS) to identify mutations that accounted for the resistance phenotype. Within resistant populations, they discovered key mutations in the promoter region of the beta-lactamase gene, ampC; in the penicillin-binding protein PBP3 which is the target of ampicillin; and in the AcrB subunit of the AcrAB-TolC efflux machinery. Importantly, mutations in the efflux machinery can impact the resistances towards other antibiotics, not just beta-lactams. To test this, they repeated the MIC experiments with other classes of antibiotics, including kanamycin, chloramphenicol, and rifampicin. Interestingly, they observed that the ∆recA strains pre-treated with ampicillin showed higher MICs towards all other antibiotic tested. This suggests that the mutations conferring resistance to ampicillin are also increasing resistance to other antibiotics.The authors then performed an impressive series of genetic, microscopy, and transcriptomic experiments to show that this increase in resistance is not driven by the SOS response, but by independent DNA repair and stress response pathways. Specifically, they show that deletion of the recA reduces the bacterium's ability to process reactive oxygen species (ROS) and repair its DNA. These factors drive accumulation of mutations that can confer resistance towards different classes of antibiotics. The conclusions are reasonably well-supported by the data, but some aspects of the data and the model need to be clarified and extended.Strengths:A major strength of the paper is the detailed bacterial genetics and transcriptomics that the authors performed to elucidate the molecular pathways responsible for this increased resistance. They systemically deleted or inactivated genes involved in the SOS response in *E. coli*. They then subjected these mutants the same MIC assays as described previously. Surprisingly, none of the other SOS gene deletions resulted an increase in drug resistance, suggesting that the SOS response is not involved in this phenotype. This led the authors to focus on the localization of DNA PolI, which also participates in DNA damage repair. Using microscopy, they discovered that in the RecA deletion background, PolI co-localizes with the bacterial chromosome at much lower rates than wild-type. This led the authors to conclude that deletion of RecA hinders PolI and DNA repair. Although the authors do not provide a mechanism, this observation is nonetheless valuable for the field and can stimulate further investigations in the future.In order to understand how RecA deletion affects cellular physiology, the authors performed RNA-seq on ampicillin-treated strains. Crucially, they discovered that in the RecA deletion strain, genes associated with antioxidative activity (cysJ, cysI, cysH, soda, sufD) and Base Excision Repair repair (mutH, mutY, mutM), which repairs oxidized forms of guanine, were all downregulated. The authors conclude that down-regulation of these genes might result in elevated levels of reactive oxygen species in the cells, which in turn, might drive the rise of resistance. Experimentally, they further demonstrated that treating the ∆recA strain with an antioxidant GSH prevents the rise of MICs. These observations will be useful for more detailed mechanistic follow-ups in the future.Weaknesses:Throughout the paper, the authors use language suggesting that ampicillin treatment of the ∆recA strain induces higher levels of mutagenesis inside the cells, leading to the rapid rise of resistance mutations. However, as the authors note, the mutants enriched by ampicillin selection can play a role in efflux and can thus change a bacterium's sensitivity to a wide range of antibiotics, in what is known as cross-resistance. The current data is not clear on whether the elevated "mutagenesis" is driven ampicillin selection or by a bona fide increase in mutation rate.Furthermore, on a technical level, the authors employed WGS to identify resistance mutations in the treated ampicillin-treated wild-type and ∆recA strains. However, the WGS methodology described in the paper is inconsistent. Notably, wild-type WGS samples were picked from non-selective plates, while ΔrecA WGS isolates were picked from selective plates with 50 μg/mL ampicillin. Such an approach biases the frequency and identity of the mutations seen in the WGS and cannot be used to support the idea that ampicillin treatment induces higher levels of mutagenesis.Finally, it is important to establish what the basal mutation rates of both the WT and ∆recA strains are. Currently, only the ampicillin-treated populations were reported. It is possible that the ∆recA strain has inherently higher mutagenesis than WT, with a larger subpopulation of resistant clones. Thus, ampicillin treatment might not in fact induce higher mutagenesis in ∆recA.Comments on revisions:Thank you for responding to the concerns raised previously. The manuscript overall has improved.

We sincerely thank the reviewer for raising this important point. In our initial submission, we acknowledge that our mutation analysis was based on a limited number of replicates (n=6), which may not have been sufficient to robustly distinguish between mutation induction and selection. In response to this concern, we have substantially expanded our experimental dataset. Specifically, we redesigned the mutation rate validation experiment by increasing the number of biological replicates in each condition to 96 independent parallel cultures. This enabled us to systematically assess mutation frequency distributions under four conditions (WT, WT+ampicillin, *ΔrecA*, *ΔrecA*+ampicillin), using both maximum likelihood estimation (MLE) and distribution-based fluctuation analysis (new Figure 1F, 1G, and Figure S5).

These expanded datasets revealed that:

(1) While the estimated mutation rate was significantly elevated in *ΔrecA*+ampicillin compared to *ΔrecA* alone (Fig. 1G),

(2) The distribution of mutation frequencies in *ΔrecA*+ampicillin was highly skewed with evident jackpot cultures (Fig. 1F), and

(3) The observed pattern significantly deviated from Poisson expectations, which is inconsistent with uniform mutagenesis and instead supports clonal selection from an early-arising mutational pool (Fig. S5).

Importantly, these new results do not contradict our original conclusions but rather extend and refine them. The previous evidence for ROS-mediated mutagenesis remains valid and is supported by our GSH experiments, transcriptomic analysis of oxidative stress genes, and DNA repair pathway repression. However, the additional data now indicate that ROS-induced variants are not uniformly induced after antibiotic exposure but are instead generated stochastically under the stress-prone *ΔrecA* background and then selectively enriched upon ampicillin treatment.

Taken together, we now propose a two-step model of resistance evolution in *ΔrecA* cells (new Figure 5):

Step i: RecA deficiency creates a hypermutable state through impaired repair and elevated ROS, increasing the probability of resistance-conferring mutations.

Step ii: β-lactam exposure acts as a selective bottleneck, enriching early-arising mutants that confer resistance.

We have revised both the Results and Discussion sections to clearly articulate this complementary relationship between mutational supply and selection, and we believe this integrated model better explains the observed phenotypes and mechanistic outcomes.

**Reviewer #2 (Public review):**
This study aims to demonstrate that *E. coli* can acquire rapid antibiotic resistance mutations in the absence of a DNA damage response. The authors employed a modified Adaptive Laboratory Evolution (ALE) workflow to investigate this, initiating the process by diluting an overnight culture 50-fold into an ampicillin selection medium. They present evidence that a recA- strain develops ampicillin resistance mutations more rapidly than the wild-type, as indicated by the Minimum Inhibitory Concentration (MIC) and mutation frequency. Whole-genome sequencing of recA- colonies resistant to ampicillin showed predominant inactivation of genes involved in the multi-drug efflux pump system, contrasting with wild-type mutations that seem to activate the chromosomal ampC cryptic promoter. Further analysis of mutants, including a lexA3 mutant incapable of inducing the SOS response, led the authors to conclude that the rapid evolution of antibiotic resistance occurs via an SOS-independent mechanism in the absence of recA. RNA sequencing suggests that antioxidative response genes drive the rapid evolution of antibiotic resistance in the recA- strain. They assert that rapid evolution is facilitated by compromised DNA repair, transcriptional repression of antioxidative stress genes, and excessive ROS accumulation.Strengths:The experiments are well-executed and the data appear reliable. It is evident that the inactivation of recA promotes faster evolutionary responses, although the exact mechanisms driving this acceleration remain elusive and deserve further investigation.Weaknesses:Some conclusions are overstated. For instance, the conclusion regarding the LexA3 allele, indicating that rapid evolution occurs in an SOS-independent manner (line 217), contradicts the introductory statement that attributes evolution to compromised DNA repair.

We thank the reviewer for this insightful observation, which highlights a central conceptual advance of our study. Our data indeed indicate that resistance evolution in *ΔrecA* occurs independently of canonical SOS induction (as shown by the lack of resistance in *lexA3*, *dpiBA*, and translesion polymerase mutants), yet is clearly associated with impaired DNA repair capacity (e.g., downregulation of *polA*, *mutH*, *mutY*).

This apparent “contradiction” reflects the dual role of RecA: it functions both as the master activator of the SOS response and as a key factor in SOS-independent repair processes. Thus, the rapid resistance evolution in *ΔrecA* is not due to loss of SOS, but rather due to the broader suppression of DNA repair pathways that RecA coordinates, which elevates mutational load under stress (This point is discussed in further detail in our response to Reviewer 1).

The claim made in the discussion of Figure 3 that the hindrance of DNA repair in recA- is crucial for rapid evolution is at best suggestive, not demonstrative. Additionally, the interpretation of the PolI data implies its role, yet it remains speculative.

We appreciate this comment and would like to respectfully clarify that our conclusion regarding the role of DNA repair impairment is supported by several independent lines of mechanistic evidence.

First, our RNA-seq analysis revealed transcriptional suppression of multiple DNA repair genes in *ΔrecA* cells following ampicillin treatment, including *polA* (DNA Pol I) and the base excision repair genes *mutH*, *mutY*, and *mutM* (Fig. 4K). This indicates that multiple repair pathways, including those responsible for correcting oxidative DNA lesions, are downregulated under these conditions.

Second, we observed a significant reduction in DNA Pol I protein expression as well as reduced colocalization with chromosomal DNA in *ΔrecA* cells, suggesting impaired engagement of repair machinery (Fig. 3C-E). These phenotypes are not limited to transcriptional signatures but extend to functional protein localization.

Third, and most importantly, resistance evolution was fully suppressed in *ΔrecA* cells upon co-treatment with glutathione (GSH), which reduces ROS levels. As GSH did not affect ampicillin killing (Fig. 4J), these findings suggest that mutagenesis and thus the emergence of resistance requires both ROS accumulation and the absence of efficient repair.

Therefore, we believe these data go beyond correlation and demonstrate a mechanistic role for DNA repair impairment in driving stress-associated resistance evolution in *ΔrecA*. We have revised the Discussion to emphasize the strength of this evidence while avoiding overstatement.

In Figure 2A table, mutations in amp promoters are leading to amino acid changes.

We thank the reviewer for spotting this inconsistency. Indeed, the *ampC* promoter mutations we identified reside in non-coding regulatory regions and do not result in amino acid substitutions. We have corrected the annotation in Fig. 2A and clarified in the main text that these mutations likely affect gene expression through transcriptional regulation, rather than protein sequence alteration.

The authors' assertion that ampicillin significantly influences persistence pathways in the wild-type strain, affecting quorum sensing, flagellar assembly, biofilm formation, and bacterial chemotaxis, lacks empirical validation.

We thank the reviewer for pointing this out. In the original version, we acknowledged transcriptional enrichment of genes related to quorum sensing, flagellar assembly, and chemotaxis in the wild-type strain upon ampicillin treatment. However, as we did not directly assess persistence phenotypes (e.g., biofilm formation or persister levels), we agree that such functional inferences were not fully supported. We have revised the relevant statements to focus solely on transcriptomic changes and have removed language suggesting direct effects on persistence pathways.

Figure 1G suggests that recA cells treated with ampicillin exhibit a strong mutator phenotype; however, it remains unclear if this can be linked to the mutations identified in Figure 2's sequencing analysis.

We appreciate the reviewer’s comment. This point is discussed in further detail in our response to Reviewer 1.

**Reviewer #3 (Public review):**
In the present work, Zhang et al investigate involvement of the bacterial DNA damage repair SOS response in the evolution of beta-lactam drug resistance evolution in *Escherichia coli*. Using a combination of microbiological, bacterial genetics, laboratory evolution, next-generation, and live-cell imaging approaches, the authors propose short-term (transient) drug resistance evolution can take place in RecA-deficient cells in an SOS response-independent manner. They propose the evolvability of drug resistance is alternatively driven by the oxidative stress imposed by accumulation of reactive oxygen species and compromised DNA repair. Overall, this is a nice study that addresses a growing and fundamental global health challenge (antimicrobial resistance).Strengths:The authors introduce new concepts to antimicrobial resistance evolution mechanisms. They show short-term exposure to beta-lactams can induce durably fixed antimicrobial resistance mutations. They propose this is due to comprised DNA repair and oxidative stress. Antibiotic resistance evolution under transient stress is poorly studied, so the authors' work is a nice mechanistic contribution to this field.Weaknesses:The authors do not show any direct evidence of altered mutation rate or accumulated DNA damage in their model.

We appreciate the reviewer’s comment. This point is discussed in further detail in our response to Reviewer 1.

**Recommendations for the authors:**

**Reviewer #1 (Recommendations for the authors):**
I would like to suggest two minor changes to the text.(1) Re. WGS data.The authors write in their response "We appreciate your concern regarding potential inconsistencies in the WGS methodology. However, we would like to clarify that the primary aim of the WGS experiment was to identify the types of mutations present in the wild type and ΔrecA strains after treatment of ampicillin, rather than to quantify or compare mutation frequencies. This purpose was explicitly stated in the manuscript.I think the source of my confusion stemmed from this part in the text:"In bacteria, resistance to most antibiotics requires the accumulation of drug resistance associated DNA mutations developed over time to provide high levels of resistance (29). To verify whether drug resistance associated DNA mutations have led to the rapid development of antibiotic resistance in recA mutant strain, we..."I would change the phrase "verify whether drug resistance associated DNA mutations have led to the rapid development of antibiotic resistance in recA mutant strain" to "identify the types of mutations present in the wild type and ΔrecA strains after treatment of ampicillin." This would explicitly state what the sequencing was for (ie. ID-ing mutations). The current phrase can give the impression that WGS was used to validate rapid or high mutagenesis.

Thanks for this suggestion. We have revised this description to “In bacteria, resistance to most antibiotics requires the accumulation of drug resistance associated DNA mutations that can arise stochastically and, under stress conditions, become enriched through selection over time to confer high levels of resistance (33). Having observed a non-random and right-skewed distribution of mutation frequencies in *ΔrecA* isolates following ampicillin exposure, we next sought to determine whether specific resistance-conferring mutations were enriched in *ΔrecA* isolates following antibiotic exposure.”

(2) Re. whether the mutations are "induced" or "pre-existing."The authors write:"We appreciate your detailed feedback on the language used to describe our data. We understand the concern regarding the use of the term "induced" in relation to beta-lactam exposure. To clarify, we employed not only beta-lactam antibiotics but also other antibiotics, such as ciprofloxacin and chloramphenicol, in our experiments (data not shown). However, we observed that beta-lactam antibiotics specifically induced the emergence of resistance or altered the MIC in our bacterial populations. If resistance had pre-existed before antibiotic exposure, we would expect other antibiotics to exhibit a similar selective effect, particularly given the potential for cross-resistance to multiple antibiotics."I think it is important to discuss the negative data for the other antibiotics (along with the other points made in your Reviewer response) in the main text.

This point is discussed in further detail in our response to Reviewer 1 (Public Review).